# Reduced production of the major allergens Bla g 1 and Bla g 2 in *Blattella germanica* after antibiotic treatment

Seogwon Lee[1], Ju Yeong Kim[1], Myung-Hee Yi[1], In-Yong Lee[1,2], Dongeun Yong[3], Tai-Soon Yong[1]*

1 Department of Environmental Medical Biology, Institute of Tropical Medicine and Arthropods of Medical Importance Resource Bank, Yonsei University College of Medicine, Seoul, Korea, 2 Convergence Research Center for Insect Vectors, College of Life Science and Bioengineering, Incheon National University, Incheon, Korea, 3 Department of Laboratory Medicine and Research Institute of Bacterial Resistance, Yonsei University College of Medicine, Seoul, Korea

* tsyong212@gmail.com

**Data Availability Statement:** All relevant data are within the manuscript and its Supporting Information files.

## Abstract

### Purpose

Allergens present in the feces or frass of cockroaches can cause allergic sensitization in humans. The use of fecal and frass extracts for immunotherapy has been previously investigated but has not yet been fully standardized. Here, we treated cockroaches with ampicillin to produce extracts with reduced amounts of total bacteria.

### Methods

We performed targeted high-throughput sequencing of 16S rDNA to compare the microbiomes of ampicillin-treated and untreated (control) cockroaches. RNA-seq was performed to identify differentially expressed genes (DEGs) in ampicillin-treated cockroaches.

### Results

Analysis of the microbiome revealed that alpha diversity was lower in the ampicillin-treated group than in the control group. Beta diversity analysis indicated that ampicillin treatment altered bacterial composition in the microbiome of cockroaches. Quantitative polymerase chain reaction revealed that almost all bacteria were removed from ampicillin-treated cockroaches. RNA-seq analysis revealed 1,236 DEGs in ampicillin-treated cockroaches (compared to untreated cockroaches). Unlike bacterial composition, the DEGs varied between the two groups. Among major allergens, the expression of *Bla g 2* decreased significantly in ampicillin-treated cockroaches (compared to untreated group).

### Conclusions

In this study, the reduced level of allergens observed in cockroaches may be related to lower amounts of total bacteria caused by treatment with antibiotics. It is possible to make a protein extract with few bacteria for use in immunotherapy.

**Funding:** Acknowledgments: This study was supported by a National Research Foundation of Korea (NRF) Grant funded by the Korean Government (MEST; Numbers NRF-2019R1A2B5B01069843 and NRF-2020R1I1A2074562).

**Competing interests:** The authors have declared that no competing interests exist.

## Introduction

Cockroaches spread pathogenic bacteria through their feces or frass while traveling between locations, such as homes, shops, and hospitals [1]. Their omnivorous nature enables them to survive under a wide variety of conditions. The German cockroach, *Blattella germanica*, and its microbiome have been extensively investigated [2–4, 6]. Different gut microbiomes of *B. germanica* were identified at various locations inside and outside laboratory settings, likely due to differences in the diets available at these locations [2]. A laboratory-based study investigating the effect of diet on *B. germanica* confirmed that their microbiome changed dynamically according to food intake [3].

Several studies have reported that antibiotics directly affect the bacterial composition in the cockroach microbiota. Rosas *et al*. showed that rifampicin altered the *B. germanica* microbiota and that the second generation of insects following antibiotic treatment underwent microbiota recovery through fecal intake [4]. Antibiotic-treated cockroaches showed changes in bacterial diversity and composition, including the removal of the endosymbiont *Blattabacterium* [4]. Another study reported difficulties in cockroach reproduction and growth following antibiotic treatment [5]. In *Riptortus pedestris*, the absence of an endosymbiont led to a decrease in hexamerin and vitellogenin, which affected egg production and insect development [6].

Cockroaches can spread pathogenic bacteria present in their gut or body to places with high human traffic, such as restaurants or hospitals [7], and the allergens in their feces or frass can cause allergen sensitization in humans [8]. Production of the major cockroach allergen Bla g 1 in female cockroaches is related to their reproductive cycle and is also affected by their food intake [9]. Bla g 1 can bind various lipids, suggesting that it has a digestive function related to the nonspecific transport of lipid molecules [10]. Similar to Bla g 1, Bla g 2 is present at high concentrations in the digestive organs of cockroaches (esophagus, gut, and proventriculus), suggesting that Bla g 2 functions as a digestive enzyme [11]. Bla g 2 is regarded the most important *B. germanica* allergen, with the rate of sensitization being the highest among common cockroach allergens at 54–71% generally [12].

Because the potency of the cockroach protein extract was different for each cohort depending on the allergen content of the extract—as recently demonstrated in several studies on allergen immunotherapy [13, 14]—it is important to select a suitable protein extract for each patient [13].

Despite these variables, no studies have been conducted to determine the effect of bacteria in the cockroach on allergen production before extracting the protein for immunotherapy. The extract of the cockroach not only contains allergens but also harbors various immunomodulatory molecules such as endotoxin and bacterial DNA from the microbiome, which are not easily removed by the filtration process. In the present study, we aimed to obtain a protein extract of *B. germanica* with reduced levels of bacteria using ampicillin, a broad-spectrum antibiotic. In addition, we attempted to investigate the amount and composition of the microbiome of cockroaches treated with ampicillin, and whether the production of allergens in the cockroach was affected by the treatment.

## Materials and methods

### Rearing conditions

Cockroaches (*B. germanica*) were reared for several generations under the same laboratory conditions to minimize the potential influence of environmental factors and diet on their performance. All cockroaches were reared in plastic boxes (27 cm × 34 cm × 19 cm) and incubated at 25°C and 50% relative humidity. *B. germanica* were fed sterilized fish food and

provided with sterilized untreated or ampicillin-containing (autoclaved before the addition of 0.025% ampicillin) tap water *ad libitum*.

## Experimental design

Newly hatched cockroaches (G1) were randomly divided into two groups (n = 5 from each group). We used G1 cockroaches because we had to administer ampicillin immediately after hatching. Group A was offered ampicillin-treated water, while group C (control) was offered untreated water. Twenty-one days after becoming adults, ampicillin-treated (A) and untreated (C) female cockroaches were sampled for further analyses. *B. germanica* were sacrificed with $CO_2$ then surface-sterilized using alcohol and flash-frozen in liquid nitrogen. They were then individually crushed using a mortar and pestle and stored at −80˚C until further analysis. The powder of the crushed body of each cockroach was used for DNA, RNA, and protein extraction. Three biological replicates were analyzed.

## DNA extraction

Total DNA was extracted using the NucleoSpin DNA Insect Kit (Macherey-Nagel, Düren, Germany) according to the manufacturer's instructions. The DNA extracted from each sample was eluted in 20 μL of elution buffer. Procedures were all conducted at a clean bench, under a sterilized hood, and in a DNA-free room. DNA concentrations were quantified using an ND-1000 Nanodrop system (Thermo-Fisher Scientific, Waltham, MA, USA).

## Next-generation sequencing

The 16S rDNA V3–V4 region was amplified through polymerase chain reaction (PCR) using forward and reverse primers (Table 1) [15, 16], in an Illumina MiSeq V3 cartridge (San Diego, CA, USA) with a 600-cycle format. A limited-cycle amplification step was performed to add multiplexing indices and Illumina sequencing adapters to the samples. The libraries were normalized, pooled, and sequenced on the Illumina MiSeq V3 cartridge platform according to the manufacturer's instructions.

## Bioinformatic and statistical analyses

Bioinformatics analyses were performed according to the methods described in our previous study [16]. Taxonomic assignment was performed using the EzBioCloud database

**Table 1. Primers used in this study.**

| Primer Name | Primer Sequence (5' → 3') |
|---|---|
| 16S rDNA V3–V4_F | TCGTCGGCAGCGTCAGATGTGTATAAGAGACAGCCTACGGGNGGCWGCAG |
| 16S rDNA V3–V4_R | GTCTCGTGGGCTCGGAGATGTGTATAAGAGACAGGACTACHVGGGTATCTAATCC |
| ActinF | CACATACAACTCCATTATGAAGTGCGA |
| ActinR | TGTCGGCAATTCCGGTACATG |
| BACT1369 | CGGTGAATACGTTCYCGG |
| PROK1492R | GGWTACCTTGTTACGACTT |
| Blag1F | CTATATGACGCCATCCGTTCTC |
| Blag1R | CACATCAACTCCCTTGTCCTT |
| Blag2F | TGATGGGAATGTACAGGTGAA A |
| Blag2R | TGTTGAGATGTCGTGAGGTTAG |
| Blag5F | GATTGATGGGAAGCAAACACAC |
| Blag5R | CGATCTCCAAGTTCTCCCAATC |

([https://www.ezbiocloud.net/](https://www.ezbiocloud.net/)) [16] and BLAST (v. 2.2.22), and pairwise alignments were generated to measure sequence similarity [17, 18]. All analyses were performed using BIOi-PLUG, a commercially available ChunLab bioinformatics cloud platform for microbiome research ([https://www.bioiplug.com/](https://www.bioiplug.com/)) [16]. The reads were normalized to 11,000 to perform the analyses. Phylogenetic analysis was performed, and Shannon indexes, unweighted pair group method with arithmetic mean (UPGMA) clustering, principal coordinates analysis (PCoA), permutational multivariate analysis of variance, linear discriminant analysis, and linear discriminant analysis effect size (LEfSe) were determined according to our previous study [16].

## Protein extraction

Total protein was extracted by first adding 2 mL of PBS to each sample. The samples were then sonicated (QSonica Q500, Fullerton, CA, USA) and centrifuged at 10,000 ×*g* for 30 min at 4˚C. The resulting supernatants were filtered using a 0.22-μm membrane filter (Millex®, Tullagreen, Carrigtwohill, Co. Cork, Ireland). Defatting was not performed as there was little fat in the sample and to minimize bacterial contamination.

## Enzyme-linked immunosorbent assay (ELISA)

Cockroach protein extracts (2 mg/mL) were diluted 100-fold to measure Bla g 1 and Bla g 2 levels and were diluted 10-fold to measure Bla g 5 level using corresponding ELISA kits (Indoor Biotechnologies, Charlottesville, VA, USA) according to the manufacturer instructions. In brief, the detection antibody and conjugate mix were used for the immunoassay, and color development was performed with the substrate 3,3′,5,5′-tetramethylbenzidine.

## RNA extraction and cDNA synthesis

Total RNA was extracted by adding 1 mL of TRIZOL Reagent (GeneAll, Seoul, Korea) to each sample. TRIZOL supernatant was added to react with the sample and was mixed with isopropanol to obtain a pellet. The RNA extracted from each sample was eluted in 20 μL of the elution buffer. A master mix comprised 5× cDNA synthesis mix, and 20× RTase was added to mRNA samples in PCR tubes for cDNA synthesis.

## Quantitative real-time PCR (qPCR)

Quantitative real-time PCR (qPCR) was performed to quantify *Bla g 1*, *Bla g 2*, *Bla g 5*, and bacterial 16S rRNA in whole cockroaches. Actin 5C (accession number AJ861721.1) was used as the internal control, and primers specific to this gene were designed for this experiment: ActinF and ActinR (Table 1) [4]. All bacterial 16S rRNA were amplified using the forward primer BACT1369 and the reverse primer PROK1492R (Table 1) from XenoTech with AMPI-GENE qPCR Mixes (ENZO, USA) [19]. *Bla g 1* (accession number EF202179.1), *Bla g 2* (accession number EF203068.1), and *Bla g 5* (accession number EF202178.1) gene expression was used as a measurement of major allergen content. We designed the following primers for this experiment: Blag1F and Blag1R, Blag2F and Blag2R, and Blag5F and Blag5R (Table 1). qPCR analyses were performed using the 2× SensiFAST™ SYBR® Hi-ROX kit (Bioline Meridian Bioscience, Humber Rd, London) with SYBR Green as the fluorescent reporter, H$_2$O, corresponding primers, and either genomic or complementary DNA. At the end of each reaction, a melting curve was generated to check the specificity of amplification and to confirm the absence of primer dimers. All reactions, including negative controls (containing water instead of DNA), were run in duplicate in 96-well plates.

## RNA-seq analysis

We used total RNA (n = 3 from each group) and the TruSeq Stranded mRNA LT Sample Prep Kit (San Diego, California, USA) to construct cDNA libraries. The protocol consisted of polyA-selected RNA extraction, RNA fragmentation, random hexamer primed reverse transcription, and 100 nt paired-end sequencing by Illumina NovaSeq 6000 (San Diego, California, USA). The libraries were quantified using qPCR according to the qPCR Quantification Protocol Guide and qualified using an Agilent Technologies 2100 Bioanalyzer.

Raw reads from the sequencer were preprocessed to remove low-quality and adapter sequences. The processed reads were aligned to the *B. germanica* genome using HISAT v2.1.0 [20]. HISAT utilizes two types of indexes for alignment (a global, whole-genome index, and tens of thousands of small local indexes). These index types are constructed using the same Burrows–Wheeler transform, and graph Ferragina Mangini index as Bowtie2. HISAT generates spliced alignments several times faster than the Burrows–Wheeler Aligner (BWA) and Bowtie because of how efficiently it utilizes these data structures and algorithms. The reference genome sequence of *B. germanica* and annotation data were downloaded from NCBI. Known transcripts were assembled using StringTie v1.3.4d [21, 22], and the results were used to calculate the expression abundance of transcripts and genes as read count or fragments per kilobase of exon per million fragments mapped value per sample. Expression profiles were used to further analyze differentially expressed genes (DEGs). DEGs or transcripts from groups with different conditions can be filtered through statistical hypothesis testing.

## Statistical analysis of gene expression

The relative abundances of gene expression were measured in the read count using StringTie. We performed statistical analyses to detect DEGs using the estimates of abundance for each gene in individual samples. Genes with more than one "zero" read count value were excluded. Filtered data were log2-transformed and subjected to trimmed mean of M-values normalization. The statistical significance of the fold change in expression (i.e., differential expression data) was determined using the exact test from edgeR [23], wherein the null hypothesis was that no difference exists among groups. The false discovery rate (FDR) was controlled by adjusting the p-value using the Benjamini-Hochberg algorithm. For DEGs, hierarchical clustering analysis was performed using complete linkage and Euclidean distance as a measure of similarity. Gene-enrichment and KEGG pathway analyses for DEGs were also performed based on the Gene Ontology (http://geneontology.org/) and KEGG pathway (https://www.genome.jp/kegg/) databases, respectively. We used the multidimensional scaling (MDS) method to visualize the similarities among samples and applied the Euclidean distance as a measure of dissimilarity. Hierarchical clustering analysis was also performed using complete linkage and Euclidean distance as a measure of similarity to display the expression patterns of differentially expressed transcripts that satisfied a |fold change| $\geq$ 2 and a raw P-value <0.05.

## Results

First-generation cockroaches reached the adult stage and were kept for an additional 21 days before being sacrificed for further analysis (Fig 1). The number of laid eggs did not vary between the groups, but the offspring were reduced approximately ten times after ampicillin treatment. In addition, no morphological differences were observed between the groups. qPCR analysis showed that the number of total bacteria in the cockroaches was 2,000 times higher in the control group than that in the ampicillin-treated group (Fig 2).

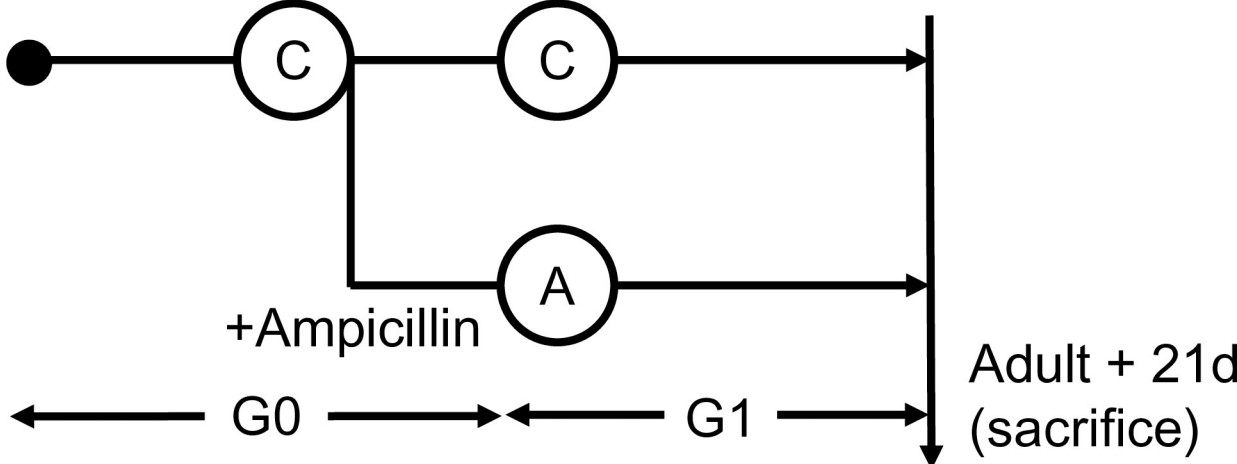

**Fig 1. Experimental design depicting ampicillin treatment of *B. germanica*.** The cockroaches were divided into two groups (A and C) and individuals were either treated with ampicillin (A) or left untreated as control specimens (C). Ampicillin was administered to cockroaches from the G1 (i.e., offspring from G0) generation, 21 days after they had reached the adult stage. Whole bodies were then collected and analyzed.

Comparison of the microbial composition data showed, in the control group, the endosymbiont *Blattabacterium* CP001487_s was the most abundant (27.43%), followed by *Fusobacterium varium*. By contrast, *Desulfovibrio*_g2 was the most abundant (64.39%) in the ampicillin-treated group (Fig 3A). LEfSe analysis of sampled cockroaches showed that, in the control group, *Blattabacterium* CP001487_s showed the greatest difference in composition, followed by *F. varium*, *Rhodopila*_uc, and *Dysgonomonas*_uc (Fig 3B). In the ampicillin-treated group, Desulfovibrio_g2 and Planctomycetes were the bacteria that showed the greatest differences in abundance (Fig 3B). Analysis of alpha diversity revealed a significantly lower number of operational taxonomic units (OTUs) (P = 0.009) in the ampicillin-treated group (Fig 4A). Although not statistically significant, phylogenetic diversity tended to be low in the ampicillin-treated group (P = 0.076) (Fig 4B), indicating low overall abundance. A significant difference was noted in the Shannon diversity index, reflecting richness and equity simultaneously (P = 0.009) (Fig 4C). Analysis of diversity using UPGMA clustering showed that the samples from the control and ampicillin-treated groups were clustered separately (S1A Fig). Similarly, principal coordinates analysis showed that both groups were clustered separately, with samples from the ampicillin-treated and control groups located on the left and right sides of the plot, respectively (S1B Fig).

RNA-sequencing was performed to explore the effect of ampicillin on gene expression in cockroaches. UPGMA clustering results showed that the control group was grouped together, but that one ampicillin-treated sample was clustered separately (Fig 5A). Principal components analysis (PCA) confirmed that separation was achieved between the control and ampicillin-treated groups (Fig 5B). Hierarchical clustering analysis between the control and ampicillin-treated groups generated a heat map of 1,236 DEGs for both groups (Fig 5C). Gene Ontology (GO) functional classification analysis revealed that these 1,236 DEGs were divided among three main categories (biological process, cellular component, and molecular function), where differential expression more than doubled in 28, 16, and 13 items, respectively, between the control and ampicillin-treated groups (Fig 6).

RNA-seq showed that the expression level of *Bla g 2* decreased by four times in the ampicillin-treated group (S1 Table). Subsequently, RNA levels of the genes encoding the three major allergens Bla g 1 (Fig 7A), Bla g 2 (Fig 7B), and Bla g 5 (Fig 7C) were measured using qPCR.

# Total bacteria

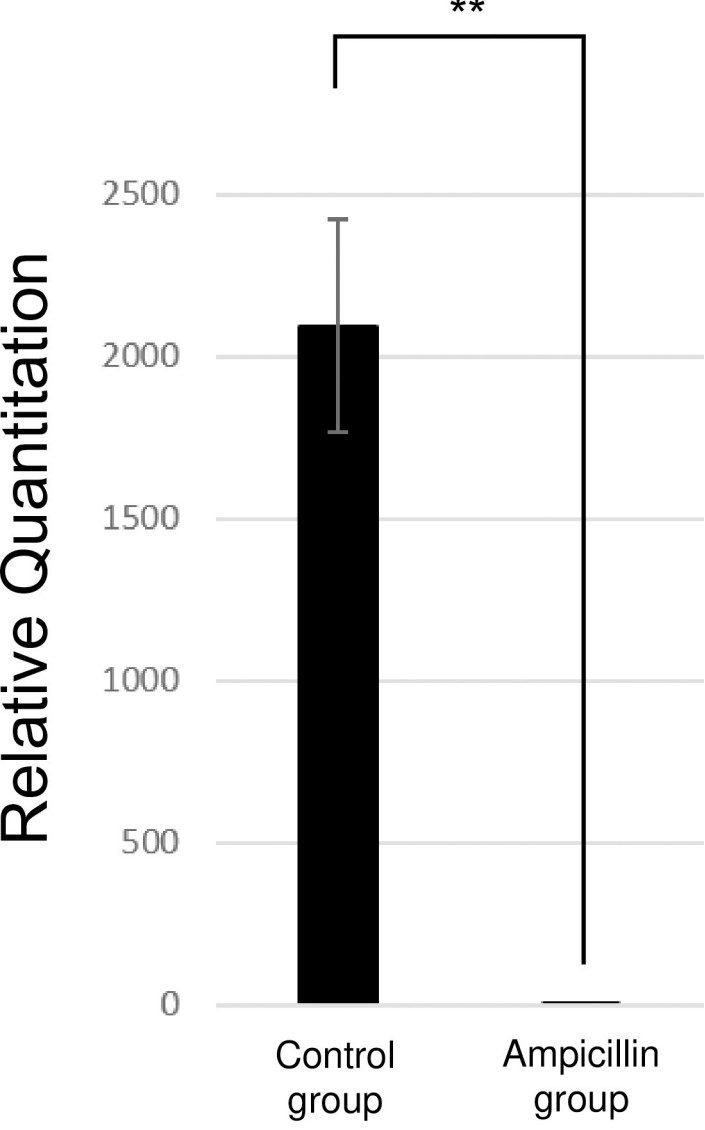

**Fig 2.** Relative quantification of the total bacterial population in the ampicillin-treated (A) and untreated (C) cockroach groups.

Our findings confirmed that the expression levels of *Bla g 1* (P = 0.000594) and *Bla g 2* (P < 0.00001), but not *Bla g 5* (P = 0.05067), were significantly decreased in the ampicillin-treated group compared to those in the control group. Additionally, we noted a larger decrease in the level of *Bla g 2* than that of *Bla g 1* (Fig 7).

At the protein level, we measured the amounts of Bla g 1 (S2A Fig), Bla g 2 (S2B Fig), and Bla g 5 (S2C Fig). The results were similar to those obtained from transcriptomic analyses. No significant difference was detected in Bla g 5 (P = 0.296897), whereas a significant decrease in the expression of Bla g 1 (P = 0.000463) and Bla g 2 (P = 0.00001) was observed in the ampicillin-treated group compared to that in the control group. Here, Bla g 2 sustained yet again a more notable decrease than Bla g 1 (S2 Fig).

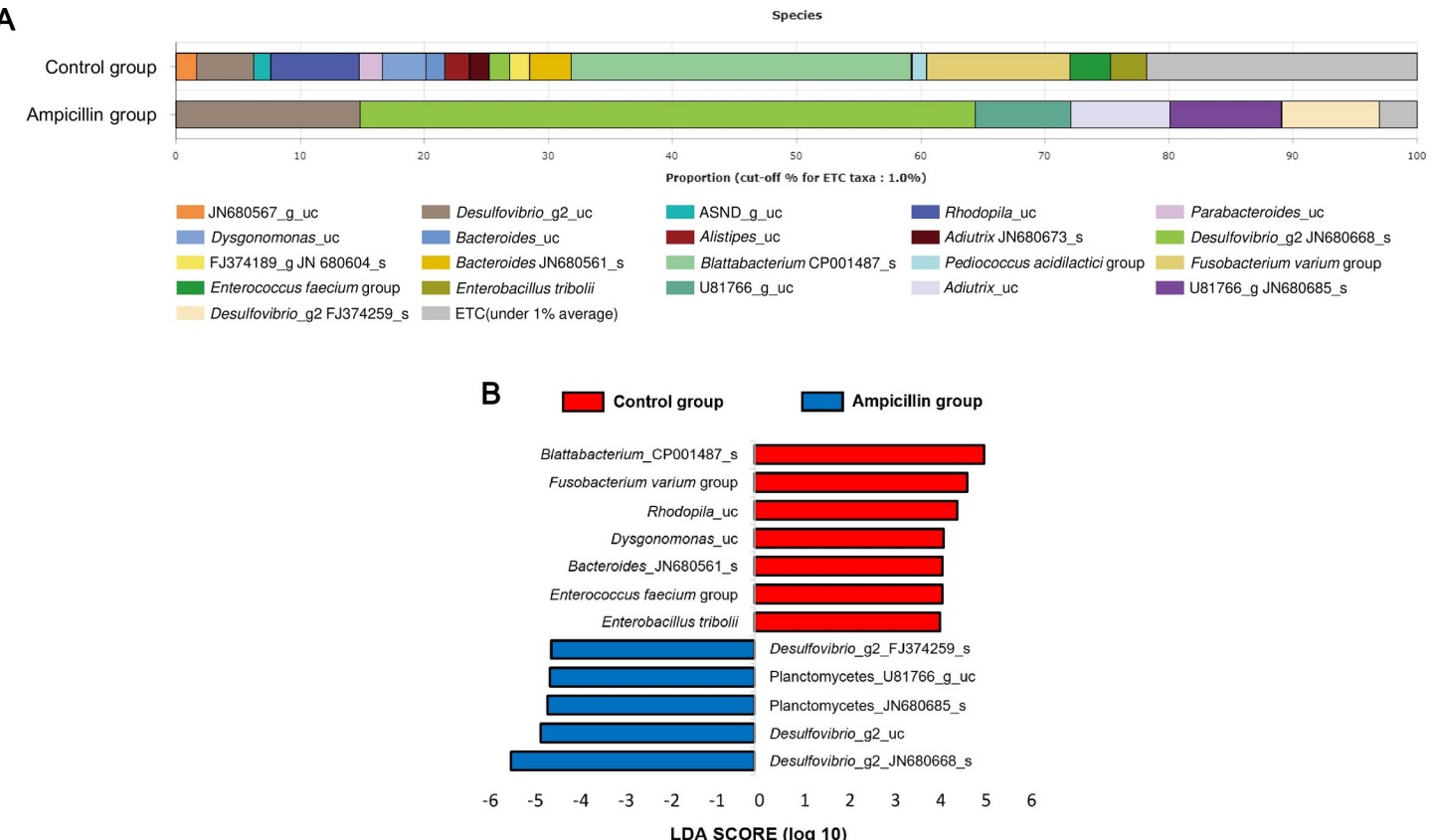

**Fig 3. Bacterial composition at the species level in the control and ampicillin-treated groups.** (A) Microbiome composition of each group (n = 5). (B) Linear discriminant analysis effect size analysis of differentially abundant bacterial taxa among the two groups. Only taxa meeting a significant (>4) linear discriminant analysis threshold are shown.

## Discussion

We treated cockroaches (*B. germanica*) with ampicillin to obtain a protein extract containing a minimal number of bacteria for immunotherapy. Analysis of *B. germanica* following treatment revealed several changes.

First, the total bacterial population was notably affected. Compared with the control group, total bacteria in cockroaches from the ampicillin-treated group disappeared almost completely, perhaps because ampicillin eliminated both gram-positive and gram-negative bacteria. One of the objectives of the study was to produce a protein extract of cockroach with a reduced bacteria content. When rearing cockroaches for clinical use such as for allergy diagnosis and immunotherapy, strict control of the bacteria using measures such as ampicillin treatment is recommended.

The microbiome study revealed marked differences at the species level. A 'super-resistant' taxon was previously identified in *B. germanica* treated with rifampicin instead of ampicillin [4]. The *Desulfovibrio* and *Planctomycetes* genera occurred in lower numbers in control cockroaches but comprised most of the microbiota of ampicillin-treated cockroaches. This finding is supported by the fact that all *Plantomycetes* are resistant to β-lactam antibiotics [24], including ampicillin, which belongs to the penicillin group of antibiotics. Similarly, *Desulfovibrio* and *Adiutrix* are resistant to ampicillin, which eliminated other bacterial species. Among the several bacterial species found in the control group, *Blattabacterium* is the most important. In

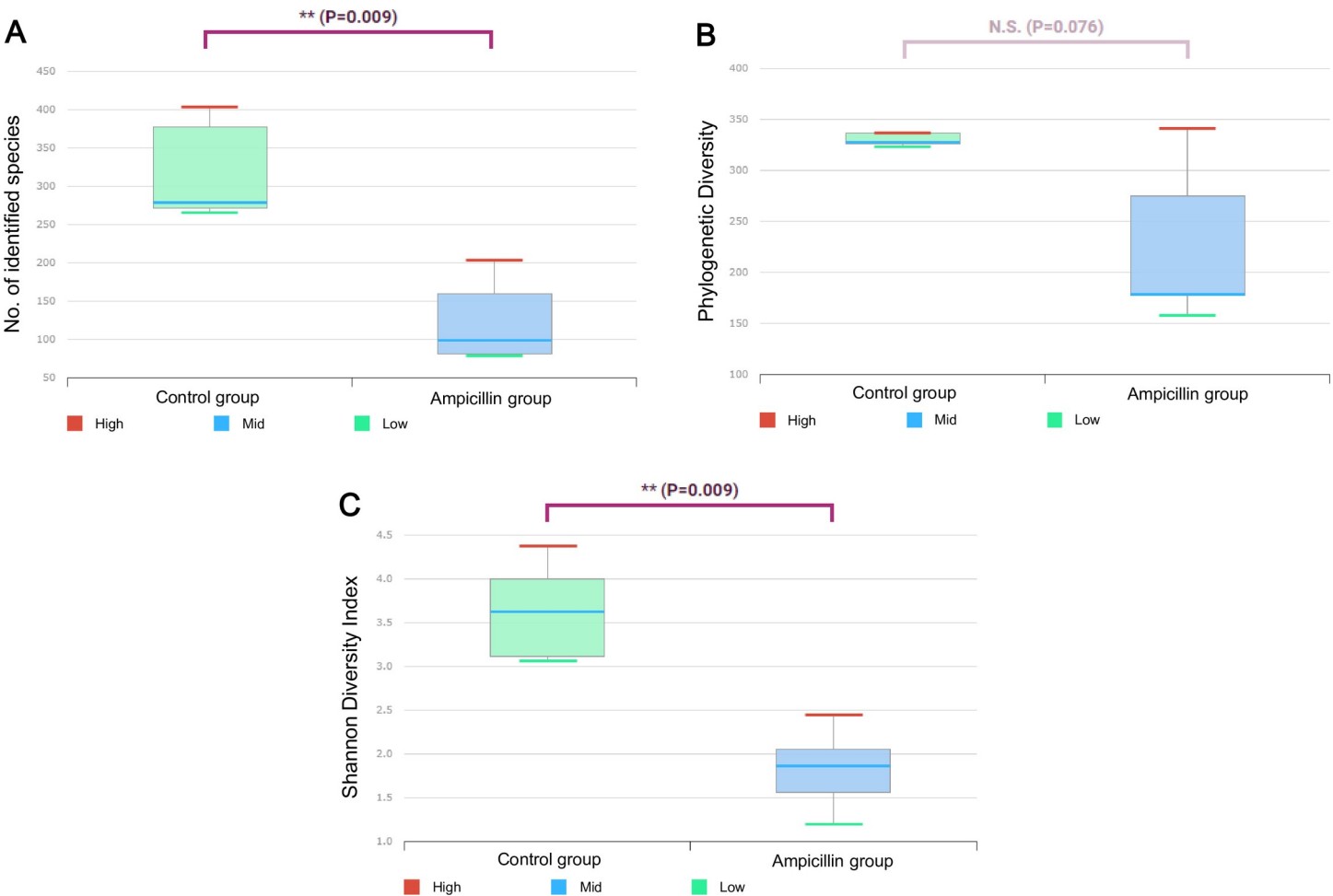

**Fig 4. Box plots showing alpha diversity in the control and ampicillin-treated groups.** (A) The number of operational taxonomic units. (B) Phylogenetic diversity (abundance). (C) Shannon diversity index (measuring richness and equity in the distribution of the species). * indicates a P-value < 0.05 from the Wilcoxon rank-sum test.

a previous study, treatment with rifampicin failed to eliminate *Blattabacterium* from the first generation of cockroaches; however, with continued treatment, the bacteria were eliminated from second-generation specimens [4]. Our data showed that treatment with ampicillin immediately eliminated *Blattabacterium* from first-generation adults. *Blattabacterium* is an endosymbiont of *B. germanica*, in which it is involved in the synthesis of essential amino acids and various vitamins, as well as in nitrogen recycling [25]. A previous study showed that tetracycline removed the endosymbiont of *Riptortus pedestris*, and that the expression of genes encoding hexamerin and vitellogenin was reduced. Consequently, these findings confirmed the factors that affected egg production and development [6]. Here, we expected that the absence of an endosymbiont would lead to several changes. Essentially, the reported decrease in bacterial composition produced differences in alpha and beta diversity. Because ampicillin reduced the bacterial load, OTUs were significantly lower in the ampicillin-treated group than in the control group. Phylogenetic diversity (indicating abundance) was not significantly different but tended to be lower in the ampicillin-treated group. The Shannon diversity index significantly decreased, reducing both richness and equity. Analysis of beta diversity using UPGMA and PCoA showed a clear difference in clustering between the two groups.

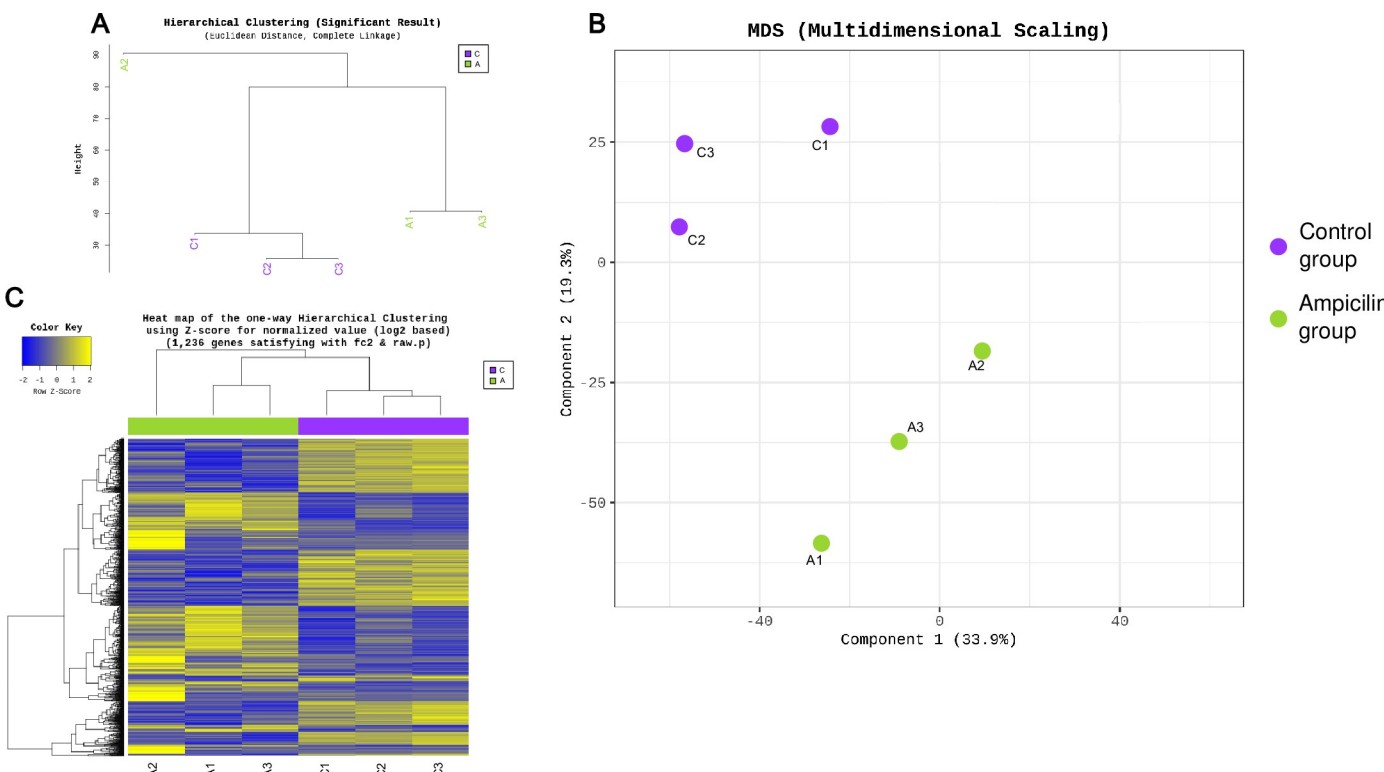

**Fig 5. Transcriptome analysis in ampicillin-treated and control cockroaches.** (A) UPGMA (unweighted pair group method with arithmetic mean) clustering. (B) Principal components analysis depicting the differences in the differentially expressed genes (DEGs) between the control and ampicillin-treated groups. (C) Heat map of transcriptional expression patterns of the two groups, displaying the expression profile of the top 1,236 DEGs for each sample in the RNA-seq dataset.

RNA-seq was performed to identify changes in gene expression at the RNA level caused by ampicillin. Results similar to those from microbiome clustering analysis were confirmed at the RNA level. Hierarchical clustering and heat map analysis showed that one of the ampicillin-treated samples was clustered separately, but samples in the control group clustered well. PCA confirmed that each group was well clustered. Most of the ampicillin-treated *B. germanica* showed decreased levels of DNA; however, gene levels were either substantially increased or decreased at the RNA level. DEGs were enriched in biological, metabolic, and cellular processes. Differential expression of various genes from the two groups was also noted in developmental process and growth, as well as in cellular component, with several differences noted between the cells and the organelles. The RNA-seq data showed that the expression level of the *Bla g 2* gene (encoding aspartic protease) was reduced by more than four times following ampicillin treatment (S1 Table). Therefore, gene expression levels for the major allergens Bla g 1, Bla g 2, and Bla g 5 were further measured via qPCR, and the protein production level was also measured.

Similar patterns of expression changes of the major allergens Bla g 1, Bla g 2, and Bla g 5 were observed at both the mRNA and protein levels. There was no significant difference in Bla g 5 expression between the ampicillin-treated cockroaches and control group, but its abundance tended to be lower in the ampicillin-treated group than in the control group. By contrast, Bla g 1 and Bla g 2 showed a significant decrease in abundance in the ampicillin-treated group compared to the control group.

The exact mechanism of allergen production in German cockroaches is unknown. However, clear differences in Bla g 1 production were observed based on the insect's level of starvation or its stage in the gonadotropic cycle [9]. Therefore, we expected to observe changes

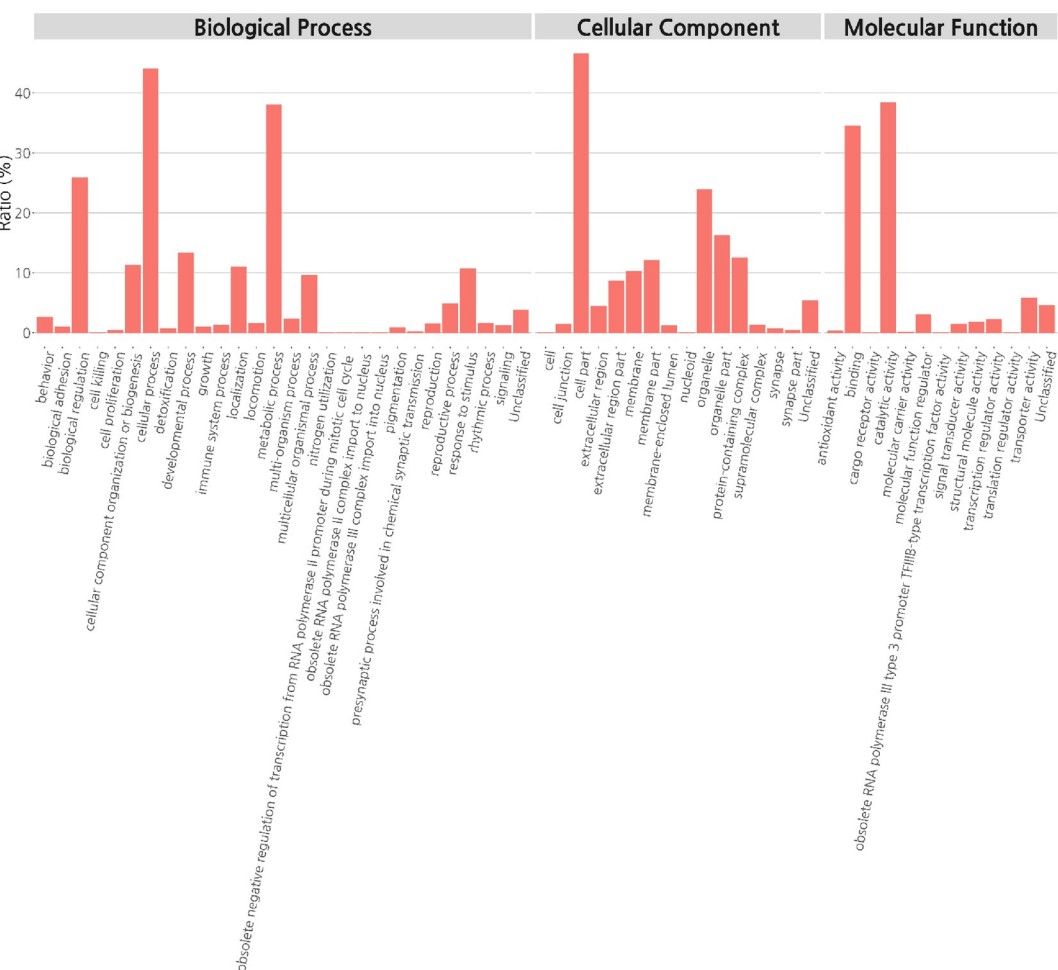

**Fig 6. Gene Ontology (GO) functional classification analysis of differentially expressed genes (DEGs) between the control and ampicillin-treated groups.** Based on sequence homology, 1,236 DEGs were categorized into three main categories, biological process, cellular component, and molecular function, with 28, 16, and 13 functional groups, respectively.

caused by several factors in the present study as well. Bacteria were removed by ampicillin treatment and likely included species that promoted the growth of cockroaches, accounting for the difference in total bacteria. The inhibition of bacterial growth may have affected allergen production, with *Blattabacterium* being probably the most influential member of the cockroach microbiota. This endosymbiont is responsible for the nitrogen cycle and the production of essential amino acids and vitamins in the German cockroach [25]. Moreover, in other insects, *Blattabacterium* reduces the expression of genes involved in reproduction and growth inhibition. Similar growth rates were observed in *R. pedestris* originally without the endosymbiont and in those treated with antibiotics to remove the bacterium [6]. German cockroaches may also experience changes in reproduction and growth due to the removal of *Blattabacterium*. RNA-seq highlighted numerous changes in gene expression. Therefore, we suggest that ampicillin may have influenced the production of allergens. Although antibiotics primarily affect bacteria, they may also indirectly affect allergens through their effects on bacteria. In this study, treatment with ampicillin eliminated *Blattabacterium*, an endosymbiont of *B. germanica*, and reduced the production of several allergens. This suggests that *Blattabacterium* may play a key role in allergen production in cockroaches either directly or indirectly. To the best

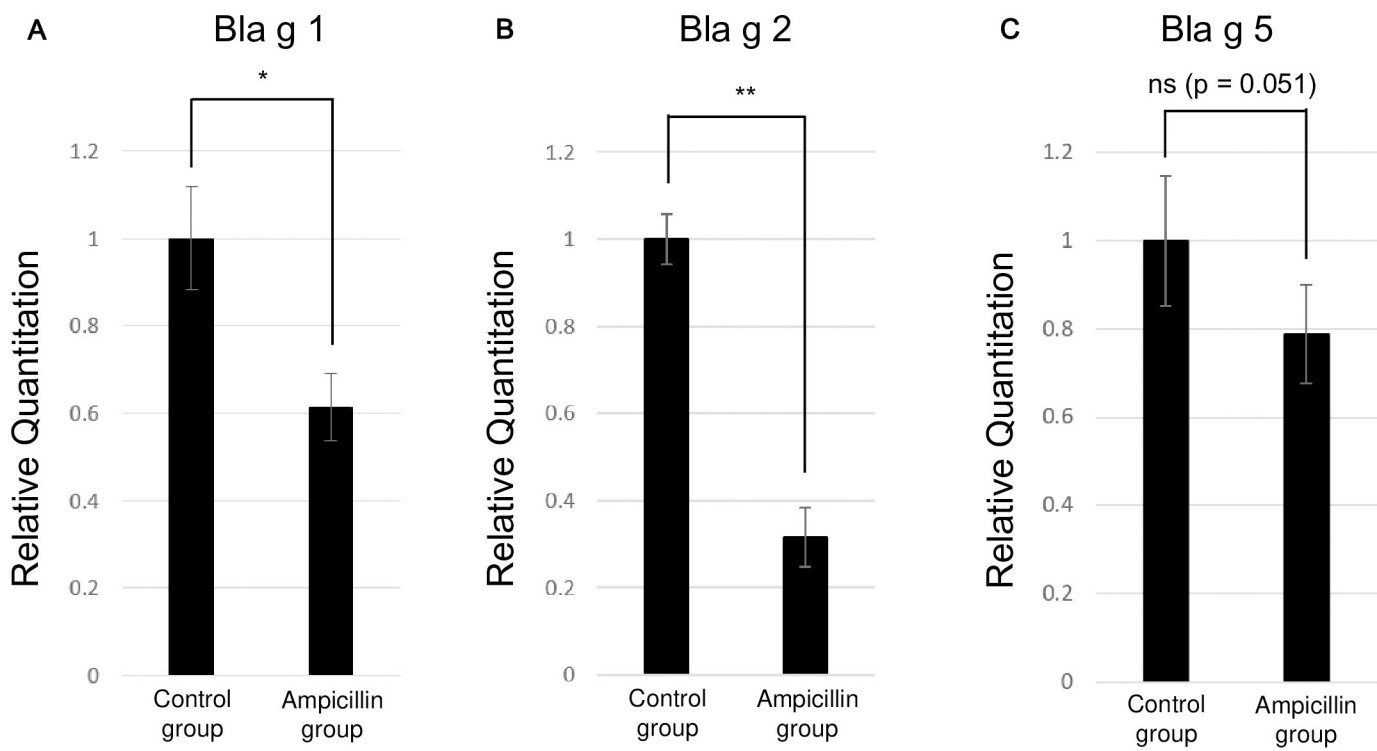

**Fig 7. Quantitative PCR (qPCR) analysis showing gene expression levels in cockroaches.** (A) *Bla g 1*, (B) *Bla g 2*, and (C) *Bla g 5*.

of our knowledge, the effect of the microbiome on cockroach allergens has not been reported. However, in *Riptortus pedestris*, the absence of the endosymbiont *Burkholderia* spp. led to a decrease in vitellogenin, an allergen in *B. germanica* [26].

A limitation of this experiment was that it was not possible to culture *Blattabacterium* alone. If *Blattabacterium* in German cockroaches could be specifically regulated, it would be possible to study only the effects of *Blattabacterium* on allergen production while excluding those of other bacteria.

In the present study, protein extraction from ampicillin-treated *B. germanica* was optimized to obtain an extract containing a small amount of Bla g 2 compared to Bla g 1 and Bla g 5, with very few bacteria. Ampicillin treatment reduced total numbers of bacteria associated with cockroaches. As a result, we suggest that reduced numbers of bacteria may have influenced the production of allergens. Future studies should investigate the effect of bacteria on the therapeutic efficacy of immunotherapy using protein extracts obtained from the German cockroach. In addition, further research is needed to confirm that a reduced allergen content of the cockroach protein extract after ampicillin treatment may induce immune tolerance in immunotherapy recipients. Furthermore, a comparative study on the effect of ampicillin treatment on the microbiome and allergen production between adult and nymph cockroaches is needed.

## Supporting information

**S1 Fig. Beta diversity in the control and ampicillin-treated groups.** (A) UPGMA (unweighted pair group method with arithmetic mean) clustering. (B) Principal coordinates analysis depicting differences in the taxonomic compositions of the bacterial communities among the two groups.
(TIF)

**S2 Fig. Allergen levels in the extracts from the two cockroach groups.** Concentrations of (A) Bla g 1, (B) Bla g 2, and (C) Bla g 5 in the extracts were measured using enzyme-linked immunosorbent assays.
(TIF)

**S1 Table. List of differentially expressed genes (DEGs) between the control and ampicillin-treated groups.**
(XLSX)

## Author Contributions

**Conceptualization:** Seogwon Lee, Ju Yeong Kim, Tai-Soon Yong.

**Data curation:** Seogwon Lee, Ju Yeong Kim, Myung-Hee Yi, Tai-Soon Yong.

**Formal analysis:** Myung-Hee Yi, Tai-Soon Yong.

**Resources:** Seogwon Lee, In-Yong Lee.

**Software:** Dongeun Yong.

**Visualization:** In-Yong Lee.

**Writing – original draft:** Seogwon Lee.

**Writing – review & editing:** Seogwon Lee, Tai-Soon Yong.

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
