## [Decision Letter · Decision Letter 0]

24 Sep 2021

PONE-D-21-26586Reduced production of the major allergen Bla g 1 and Bla g 2 in Blattella germanica after antibiotic treatmentPLOS ONE

Dear Dr. Yong,

Thank you for submitting your manuscript to PLOS ONE. After careful consideration, we feel that it has merit but does not fully meet PLOS ONE’s publication criteria as it currently stands. Therefore, we invite you to submit a revised version of the manuscript that addresses the points raised during the review process.

We look forward to receiving your revised manuscript.

Kind regards,

Chun Wie Chong

Academic Editor

PLOS ONE

Journal Requirements:

[This study was supported by a National Research Foundation of Korea (NRF) Grant funded by the Korean Government (MEST; Numbers NRF-2019R1A2B5B01069843 and NRF-2020R1I1A2074562).]

 [Acknowledgments:

This study was supported by a National Research Foundation of Korea (NRF) Grant funded by the Korean Government (MEST; Numbers NRF-2019R1A2B5B01069843 and NRF-2020R1I1A2074562).]

3. PLOS requires an ORCID iD for the corresponding author in Editorial Manager on papers submitted after December 6th, 2016. Please ensure that you have an ORCID iD and that it is validated in Editorial Manager. To do this, go to ‘Update my Information’ (in the upper left-hand corner of the main menu), and click on the Fetch/Validate link next to the ORCID field. This will take you to the ORCID site and allow you to create a new iD or authenticate a pre-existing iD in Editorial Manager. Please see the following video for instructions on linking an ORCID iD to your Editorial Manager account: https://www.youtube.com/watch?v=_xcclfuvtxQ.

Reviewers' comments:

Reviewer's Responses to Questions

**Comments to the Author**

1. Is the manuscript technically sound, and do the data support the conclusions?

Reviewer #1: Yes

Reviewer #2: Yes

2. Has the statistical analysis been performed appropriately and rigorously? 

Reviewer #1: Yes

Reviewer #2: Yes

3. Have the authors made all data underlying the findings in their manuscript fully available?

Reviewer #1: Yes

Reviewer #2: No

4. Is the manuscript presented in an intelligible fashion and written in standard English?

Reviewer #1: Yes

Reviewer #2: Yes

5. Review Comments to the Author

Reviewer #1: The manuscript prepared by Lee and team reported interesting findings that ampicillin-treated cockroaches had significantly reduced total bacteria or gut microbiome which eventually contributed to lower Bla g 1 and 2 allergen levels. The manuscript was organised in an easy-to-follow manner and very limited syntax errors. All the data were analysed and presented systematically.

The manuscript can be considered for publication at PloS One after few minor amendments. The aim of the reported study was to obtain extract of B. germanica with reduced levels of bacteria/allergen for immunotherapy. Wonder the reduced allergenicity (reduced Bla g 1 and 2 allergens) after ampicillin treatment be able to trigger immune recognition/stimulation of the recipient of the immunotherapy? Has this been verified?

How many biological replicates were conducted?

How about the allergenicity and concentrations of other allergens (besides Bla g 1, 2 and 5 allergens)? Any clues? How significant are other allergens on allergenicity and/or immunotherapy? Shall this be verified or take into consideration? Suggest providing some background summary of other allergens also.

How do you assign the newly hatched cockroaches into control and treatment group? How do you determine the sex (male/female) of the newly hatched?

What is the rationale of using G1 and female cockroaches for this experiment? Will the cockroaches treated with ampicillin be able to produce eggs? Are there any morphological and development changes to the ampicillin-treated cockroaches after endosymbiont/microbiome changes?

Will the overgrowth of the ampicillin-resistant bacteria affect the allergenicity of the cockroaches?

Were the five cockroaches use for the DNA extraction female? How many samples were used for next-generation sequencing? All five, three or pool of 5?

With reference to your method on protein extraction (lines 144-147), wonder will sonication be sufficient to solubilise/release most of the proteins of the cockroach?

Please specify clearly where the total RNA was extracted from. Was it from the same five cockroaches that were used for DNA extraction?

Another aspect that you highlighted is the standardisation of allergen for immunotherapy, how would you standardise your protein extract? Which generation of the ampicillin treated cockroaches would you recommend? Any feasibility or practically analyses?

The authors may consider revising and standardising the reference format and abbreviation (e.g. PCA / PCoA) used within the article and also according to the journal’s specification/requirement.

Reviewer #2: The authors analysed and compared between the microbiomes of the German cockroach (Blattella germanica) treated with ampicillin and the untreated control. The aim for the antibiotic treatment was to reduce the total bacterial population that may directly or indirectly be involved in allergen production found in the cockroach. This would assist in producing protein (allergen) extracts with minimal presence of the bacteria for immunotherapy.

Overall, the manuscript was well-written. The microbiome data analyses based on 16S rRNA amplicon sequencing and RNA-seq were thorough. An adequate number of samples and statistics were also included. However, whilst the manuscript is technically sound, there are several issues that I would like to highlight to the authors. Addressing these comments by the authors would hopefully help to strengthen their manuscript. The comments are categorised under major, minor and miscellaneous as follows:

Major comments

1. In the introduction, the authors stressed that resident bacteria play a role in the insect’s growth and reproduction, and antibiotic treatments alter its microbiome composition. The authors then presented their results that supported the latter. They showed that antibiotics brought about several changes to the bacterial population, microbiome composition, gene expression and allergen (bla protein) quantification, etc. These are all justified.

However, the underlying motivation of this study (i.e. through antibiotic treatment of B. germanica), as stated by the authors (in the abstract, introduction and discussion), was to obtain the allergen protein extracts with “minimal amount of bacteria” for immunotherapy purposes. The authors brought this argument only towards the end of the introduction (lines 81-88), and mentioned this aim very briefly in the discussion (lines 320-321) without providing a strong justification for the antibiotic treatment. I wondered if filter sterilising the crude extracts (line 146) would solve the problem i.e. to obtain the protein extracts without bacteria and without needing the antibiotic application. If filter-sterilising the extracts do not work, why not?

In addition, the justification/rationale for the microbiome analyses were weak and were not even highlighted at the end of the introduction despite these analyses being the major results of this study. What questions were the authors attempting to answer by performing the microbiome analyses? What was the rationale?

The authors need to address the above gaps by providing and elaborating on the reasoning behind the study and analyses presented. The justification should be clarified in the introduction and re-stated in the discussion, and should be linked to the results in a meaningful way. Doing so serves as a fundamental basis to any scientific study including this one.

2. The discussion section, although in general was well-written, lacked the depth necessary to understand the results in a more meaningful way. Most of what was written in the discussion is a reiteration of the results. For example, it is clear that the antibiotic treatment gave rise to ‘super-resistant’ taxa (Desulfovibrio and Plantomycetes) in/on the cockroach, and the control cockroach contained Blattabacterium as the main taxon (lines 327-354). However, how does these results relate to the Bla protein production quantified in the study or other insights that might be derived from the results? The authors could extend this discussion by e.g. stating that it is probably Blattabacterium that may be involved in the allergen production by the cockroach directly/indirectly and provide evidence based on the other results and those of past studies related to the study’s main objective.

The arguments presented for the RNA-seq results (lines 355-368) were also equally weak. The authors found that there was a “significant difference in the molecular functions related to catalytic activity and binding” (line 366) but did not link these to anything relevant in this study. Gene expression of the microbiome was found higher in antibiotic-treated cockroach compared to control, but can the authors speculate why this was so?

The authors discussed that ampicillin may have influenced the production of allergens directly or indirectly (lines 390-392), which I agree, based on their results. However, would it be possible to do some correlation analyses between the results the authors obtained to support this statement?

The authors went on by stating that a limitation of the study was that Blattabacterium is not yet culturable. However, there are methods that the authors can use e.g. qPCR, FISH, to study the bacterium’s behaviour against the bacterial population. These can be explored and mentioned.

Minor comments

1. Some of the results, e.g. Figs. 5 and 9, only re-iterated the message of the earlier figure(s) and thus could be better presented as supplementary data. This ensures that the section is more concise.

2. In addition to the results, there are also data that the authors should provide as supplementary. These were mentioned as ‘data not shown’ (lines 99, 296). Having these data available are in line with PLOS ONE’s principles on data transparency, availability and reproducibility.

3. Some of the methods need to be a bit more elaborated, particularly on ELISA (lines 150-151) and TRIZOL RNA extraction (Line 154).

4. Some sub-sections in the methods need to be condensed, re-organised and have its format readjusted. Repetition was found for sample processing (Line 107-112), RNA extraction/cDNA libraries (Lines 153-157; Lines 178-184), statistical analyses (Lines 130-141; lines >200). Reorganisation/merging the sub-sections would make it less repetitive and ensure that readers can follow the procedures better.

Miscellaneous

Line 62: Reference to the author is missing

Line 82: What do the authors mean by “suitable” protein extract? What is considered “suitable”? This comment links to the first major comment of this review.

Line 85-86: The effect of cockroach on what? The same also goes for line 395: ‘the effects of Blattabacterium” on what? And line 402: To investigate the effect of bacteria on patients in what sense? These need more clarification.

Line 99: Add ‘the’ before ‘concentration’

Line 117-118: Format adjustment required

Line 127: Table caption should be placed before the table

Line 130-131: “Bioinformatic analyses”, not “bioinformatics analyses”

Line 279-280: The authors mentioned that the 1236 DEGs doubled, but which one doubled? Was it in ampicillin-treated group? Please clarify.

Line 295-296: “The expression of Bla g 2 RNA more than doubled in the ampicillin-treated group”. Isn’t this ironic compared to the results presented in Figs. 8 and 9? What is the difference between this result not shown and the results presented in Fig. 8? Aren’t they both RNA-seq of the bla gene?

Line 352: Ratio of bacteria to what?

Line 358-359: What did the authors mean by “clustering occurred first among the control group”?

Line 369-370: This opening statement is confusing when the authors stated “…to confirm changes in the allergens according to the changes”. Please rephrase this.

6. PLOS authors have the option to publish the peer review history of their article (what does this mean?). If published, this will include your full peer review and any attached files.

Reviewer #1: No

Reviewer #2: No

---

## [Author Response · Author response to Decision Letter 0]

25 Oct 2021

Oct 21, 2021

Dr Emily Chenette

Editor-in-Chief

PLoS One

Re: #PONE-D-21-26586; “Reduced production of the major allergens Bla g 1 and Bla g 2 in Blattella germanica after antibiotic treatment”

Dear Editors and Reviewers:

Thank you for your helpful comments regarding our manuscript titled “Reduced production of the major allergens Bla g 1 and Bla g 2 in Blattella germanica after antibiotic treatment” (#PONE-D-21-26586) and for providing us with the opportunity to submit a revised version. We have carefully revised the manuscript to reflect the comments of the reviewers, addressing all issues and questions raised. We believe that the quality of this manuscript has been greatly improved thanks to your invaluable comments. Once again, thank you very much for your consideration and effort. Please find the point-by-point responses to each of the reviewers’ comments attached. We hope that the manuscript will now be deemed suitable for publication in PLoS One.

I look forward to hearing from you.

Sincerely,

Tai-Soon Yong

Department of Environmental Medical Biology, Institute of Tropical Medicine and

Arthropods of Medical Importance Resource Bank, Yonsei University College of Medicine

Seoul 03722, Korea

Telephone: +82-2-2228-1841

Fax: +82-2-363-8676

E-mail: tsyong212@gmail.com 

Responses to Reviewer Comments

Reviewer #1: The aim of the reported study was to obtain extract of B. germanica with reduced levels of bacteria/allergen for immunotherapy. Wonder the reduced allergenicity (reduced Bla g 1 and 2 allergens) after ampicillin treatment be able to trigger immune recognition/stimulation of the recipient of the immunotherapy? Has this been verified?

-> Thank you for raising this important comment. This study did not demonstrate that reduction in allergens can trigger immune recognition/stimulation of the recipient of the immunotherapy. To address this point, we have added the following statement to the Discussion that further research on this aspect is needed:

(Lines 417–419, in DISCUSSION) In addition, further research is needed to confirm that a reduced allergen content of the cockroach protein extract after ampicillin treatment may induce immune tolerance in immunotherapy recipients.

How many biological replicates were conducted? 

-> We have added the following sentence to the Materials and Methods to clarify this part of the experimental design:

(Lines 112–113, in MATERIALS AND METHODS) Three biological replicates were analyzed.

How about the allergenicity and concentrations of other allergens (besides Bla g 1, 2 and 5 allergens)? Any clues? How significant are other allergens on allergenicity and/or immunotherapy? Shall this be verified or take into consideration? Suggest providing some background summary of other allergens also.

-> We only measured the Bla g 1, Bla g 2, and Bla g 5 allergens in this study. These were selected as they are the major allergens that cause allergic sensitization and commonly result in symptoms. We agree that it would have been ideal to have measured other minor allergens, but unfortunately we did not have sufficient RNA and protein samples to measure all allergens. In addition, there are only commercial kits available for the German cockroach allergens Bla g 1, Bla g 2, and Bla g 5. We have added the following sentence to the Discussion to justify this choice:

(Lines 377–378, in DISCUSSION) Gene expression levels for the major allergens Bla g 1, Bla g 2, and Bla g 5 were further measured… 

How do you assign the newly hatched cockroaches into control and treatment group? How do you determine the sex (male/female) of the newly hatched?

-> Newly hatched cockroaches were randomly divided into the two groups. After breeding with a male cockroach and rearing the newly hatched cockroaches until reaching the adult stage, only females were used in the experiment. We have added the word “randomly” in the following sentence of the Materials and Methods to clarify this point:

(Lines 105–106, in MATERIALS AND METHODS) Newly hatched cockroaches (G1) were randomly divided into two groups.

What is the rationale of using G1 and female cockroaches for this experiment? 

-> We used G1 cockroaches because we had to administer the ampicillin treatment immediately after hatching. Although not described in this manuscript, female cockroaches were used for an initial reproduction study.

Will the cockroaches treated with ampicillin be able to produce eggs? 

-> The cockroaches laid eggs, but few offspring were born. Therefore, we are currently focusing on this issue to determine the reasons for the impact on fertility. 

Are there any morphological and development changes to the ampicillin-treated cockroaches after endosymbiont/microbiome changes?

-> There was no statistically significant difference in the development rate or size between groups in our study.

Will the overgrowth of the ampicillin-resistant bacteria affect the allergenicity of the cockroaches?

-> Since the total amount of bacteria was greatly reduced by the treatment, it appears unlikely that the treatment resulted in ampicillin-resistant bacteria overgrowth.

Were the five cockroaches use for the DNA extraction female? How many samples were used for next-generation sequencing? All five, three or pool of 5?

-> The powder of the crushed body of each cockroach was used for DNA, RNA, and protein extraction. Five DNA samples were used in the microbiome study. For RNAseq, 3 out of the 5 RNA samples were used. We have added the following to the Material and Methods to clarify:

(Line 185, in MATERIALS AND METHODS) (n = 3 from each group)

With reference to your method on protein extraction (lines 144-147), wonder will sonication be sufficient to solubilise/release most of the proteins of the cockroach?

-> As mentioned above, the powder of the crushed body of each cockroach was used for DNA, RNA, and protein extraction. Since only one-third of each cockroach was used for protein extraction, there was almost no fat in the sample and therefore there was no need to remove fat. In addition, since this was a microbiome-sensitive study, the protein manufacturing process was minimized to reduce potential contamination.

Please specify clearly where the total RNA was extracted from. Was it from the same five cockroaches that were used for DNA extraction?

-> We have added the following sentence to clarify the sampling methods:

(Lines 111–112, in MATERIALS AND METHODS) The powder of the crushed body of each cockroach was used for DNA, RNA, and protein extraction.

Another aspect that you highlighted is the standardisation of allergen for immunotherapy, how would you standardise your protein extract?

-> In this study, we propose a method for producing a cockroach extract using ampicillin. This method can help to reduce the amount of bacteria compared to the conventional method, and it was confirmed that the amount of allergen is also changed by this treatment. In addition, we have added the following statement to the Discussion that additional studies are needed to apply this method for producing an immunotherapeutic agent.

(Lines 417–419, in DISCUSSION) In addition, further research is needed to confirm that a reduced allergen content of the cockroach protein extract after ampicillin treatment may induce immune tolerance in immunotherapy recipients.

Which generation of the ampicillin treated cockroaches would you recommend? Any feasibility or practically analyses?

-> We treated the cockroaches with antibiotics as soon as they hatched. In the future, we plan to study the effects of antibiotic treatment at the adult stage.

The authors may consider revising and standardising the reference format and abbreviation (e.g. PCA / PCoA) used within the article and also according to the journal’s specification/requirement.

-> We have revised abbreviation (e.g. PCA / PCoA) used within the article. In addition, references were carefully checked to meet the journal’s standards

 

Reviewer #2: The authors analysed and compared between the microbiomes of the German cockroach (Blattella germanica) treated with ampicillin and the untreated control. The aim for the antibiotic treatment was to reduce the total bacterial population that may directly or indirectly be involved in allergen production found in the cockroach. This would assist in producing protein (allergen) extracts with minimal presence of the bacteria for immunotherapy. 

Overall, the manuscript was well-written. The microbiome data analyses based on 16S rRNA amplicon sequencing and RNA-seq were thorough. An adequate number of samples and statistics were also included. However, whilst the manuscript is technically sound, there are several issues that I would like to highlight to the authors. Addressing these comments by the authors would hopefully help to strengthen their manuscript. The comments are categorised under major, minor and miscellaneous as follows:

Major comments

1. In the introduction, the authors stressed that resident bacteria play a role in the insect’s growth and reproduction, and antibiotic treatments alter its microbiome composition. The authors then presented their results that supported the latter. They showed that antibiotics brought about several changes to the bacterial population, microbiome composition, gene expression and allergen (bla protein) quantification, etc. These are all justified. 

However, the underlying motivation of this study (i.e. through antibiotic treatment of B. germanica), as stated by the authors (in the abstract, introduction and discussion), was to obtain the allergen protein extracts with “minimal amount of bacteria” for immunotherapy purposes. The authors brought this argument only towards the end of the introduction (lines 81-88), and mentioned this aim very briefly in the discussion (lines 320-321) without providing a strong justification for the antibiotic treatment. I wondered if filter sterilising the crude extracts (line 146) would solve the problem i.e. to obtain the protein extracts without bacteria and without needing the antibiotic application. If filter-sterilising the extracts do not work, why not? 

In addition, the justification/rationale for the microbiome analyses were weak and were not even highlighted at the end of the introduction despite these analyses being the major results of this study. What questions were the authors attempting to answer by performing the microbiome analyses? What was the rationale?

The authors need to address the above gaps by providing and elaborating on the reasoning behind the study and analyses presented. The justification should be clarified in the introduction and re-stated in the discussion, and should be linked to the results in a meaningful way. Doing so serves as a fundamental basis to any scientific study including this one.

-> Thank you very much for the comment and advice. Based on your comments, the following sentences have been added. We are grateful that you raised these issues, which has helped us to strengthen the rationale of the study in the manuscript. With regard to the method, we filtered live bacteria so that they were not included in the final protein extract. However, substances produced by bacteria such as lipopolysaccharide and bacterial DNA can pass through the filter. Nevertheless, antibiotics can decrease these as well.

(Lines 86–88, in INTRODUCTION) The extract of the cockroach not only contains allergens but also harbors various immunomodulatory molecules such as endotoxin and bacterial DNA from the microbiome, which are not easily removed by the filtration process.

(Lines 90–93, in INTRODUCTION) In addition, we attempted to investigate the amount and composition of the microbiome of cockroaches treated with ampicillin, and whether the production of allergens in the cockroach was affected by the treatment.

(Lines 335–338, in DISCUSSION) One of the objectives of the study was to produce a protein extract of cockroach with a reduced bacteria content. When rearing cockroaches for clinical use such as for allergy diagnosis and immunotherapy, strict control of the bacteria using measures such as ampicillin treatment is recommended.

2. The discussion section, although in general was well-written, lacked the depth necessary to understand the results in a more meaningful way. Most of what was written in the discussion is a reiteration of the results. For example, it is clear that the antibiotic treatment gave rise to ‘super-resistant’ taxa (Desulfovibrio and Plantomycetes) in/on the cockroach, and the control cockroach contained Blattabacterium as the main taxon (lines 327-354). However, how does these results relate to the Bla protein production quantified in the study or other insights that might be derived from the results? The authors could extend this discussion by e.g. stating that it is probably Blattabacterium that may be involved in the allergen production by the cockroach directly/indirectly and provide evidence based on the other results and those of past studies related to the study’s main objective.

According to your comments, we have added the following sentences to the Discussion: 

(Lines 403–406, in DISCUSSION) In this study, treatment with ampicillin eliminated Blattabacterium, an endosymbiont of B. germanica, and reduced the production of several allergens. This suggests that Blattabacterium may play a key role in allergen production in cockroaches either directly or indirectly.

The arguments presented for the RNA-seq results (lines 355-368) were also equally weak. The authors found that there was a “significant difference in the molecular functions related to catalytic activity and binding” (line 366) but did not link these to anything relevant in this study. Gene expression of the microbiome was found higher in antibiotic-treated cockroach compared to control, but can the authors speculate why this was so?

-> We agree with your opinion. The content related to “catalytic activity and binding” was removed because it did not seem to be directly relevant to the focus of the paper and therefore impacted flow. It seems that the sentence related to “microbiome’s gene expression” was also misunderstood. This sentence has also been removed. Alternatively, to provide clearer interpretation of the RNAseq data, we have added a section related to allergen (Bla g 2) production along with supplementary data (S1 Table):

(Lines 375–379, in DISCUSSION) The RNA-seq data showed that the expression level of the Bla g 2 gene (encoding aspartic protease) was reduced by more than four times following ampicillin treatment (S1 Table). Therefore, gene expression levels for the major allergens Bla g 1, Bla g 2, and Bla g 5 were further measured via qPCR, and the protein production level was also measured.

The authors discussed that ampicillin may have influenced the production of allergens directly or indirectly (lines 390-392), which I agree, based on their results. However, would it be possible to do some correlation analyses between the results the authors obtained to support this statement?

Based on the results obtained at this point, we can speculate that the production of the allergen decreased after ampicillin treatment due to a reduction of the microbiome, including reduced abundance of Blattabacterium. 

The authors went on by stating that a limitation of the study was that Blattabacterium is not yet culturable. However, there are methods that the authors can use e.g. qPCR, FISH, to study the bacterium’s behaviour against the bacterial population. These can be explored and mentioned.

-> Our intention with this statement was not regarding the measurement of the bacterial population but rather to gain control of the increase or decrease of the Blattabacterium population. We have corrected the sentence as follows for clarification:

(Lines 408–409, in DISCUSSION) “… specifically targeted…”, � “… specifically regulated…”,

Minor comments

1. Some of the results, e.g. Figs. 5 and 9, only re-iterated the message of the earlier figure(s) and thus could be better presented as supplementary data. This ensures that the section is more concise.

-> Based on your comments, we have moved Figs. 5 and 9 to the supplementary data.

2. In addition to the results, there are also data that the authors should provide as supplementary. These were mentioned as ‘data not shown’ (lines 99, 296). Having these data available are in line with PLOS ONE’s principles on data transparency, availability and reproducibility.

-> We have removed the following sentence:

(Line 102, in MATERIALS AND METHODS) “The concentration of ampicillin was set based on data from a preliminary study, which showed that the concentration did not significantly affect the survival of B. germanica (data not shown).”

-> The following sentence has been revised and a supplementary table has been added. 

(Lines 303–304) “RNA-seq showed that the expression level of Bla g 2 decreased by four times in the ampicillin-treated group (S1 Table).”

3. Some of the methods need to be a bit more elaborated, particularly on ELISA (lines 150-151) and TRIZOL RNA extraction (Line 154).

-> We have modified or added the following sentences to the Materials and Methods.

(Lines 152–157, MATERIALS AND METHODS) Cockroach protein extracts (2 mg/mL) were diluted 100-fold to measure Bla g 1 and Bla g 2 levels and were diluted 10-fold to measure Bla g 5 level using corresponding ELISA kits (Indoor Biotechnologies, Charlottesville, VA, USA) according to the manufacturer instructions. In brief, the detection antibody and conjugate mix were used for the immunoassay, and color development was performed with the substrate 3,3′,5,5′-tetramethylbenzidine.

(Lines 161–162, MATERIALS AND METHODS) TRIZOL supernatant was added to react with the sample and was mixed with isopropanol to obtain a pellet.

4. Some sub-sections in the methods need to be condensed, re-organised and have its format readjusted. Repetition was found for sample processing (Line 107-112), RNA extraction/cDNA libraries (Lines 153-157; Lines 178-184), statistical analyses (Lines 130-141; lines >200). Reorganisation/merging the sub-sections would make it less repetitive and ensure that readers can follow the procedures better.

We have closely reviewed the structure of the Methods section and have removed the following duplicated sentence:

(Line 116, in MATERIALS AND METHODS) Cockroaches (n = 5 from each group) were frozen in liquid nitrogen and individually crushed using a mortar and pestle.

The sample processing details for the microbiome and RNA-seq analyses were not removed given the different statistical processing methods.

Miscellaneous

Line 62: Reference to the author is missing

-> We have added the reference to Line 63.

Line 82: What do the authors mean by “suitable” protein extract? What is considered “suitable”? This comment links to the first major comment of this review.

-> We hope that we have sufficiently responded to your first major comment. In “Cockroach Extract Composition Greatly Impacts T Cell Potency in Cockroach-Allergic Donors [13]” (attached), the degree of the patient’s allergen sensitivity depended on the allergen content in the extract. This was the basis of the phrase “suitable protein extract,” which refers to generating extracts of different allergen contents tailored for each individual.

Line 85-86: The effect of cockroach on what? The same also goes for line 395: ‘the effects of Blattabacterium” on what? And line 402: To investigate the effect of bacteria on patients in what sense? These need more clarification.

-> We have clarified the indicated sentences as follows:

(Lines 84–86, in INTRODUCTION) Despite these variables, no studies have been conducted to determine the effect of bacteria in the cockroach on allergen production before extracting the protein for immunotherapy.

(Lines 409–410, in DISCUSSION) the effects of Blattabacterium on allergen production

(Lines 415–417, in DISCUSSION) Future studies should investigate the effect of bacteria on the therapeutic efficacy of immunotherapy using protein extracts obtained from the German cockroach.

Line 117-118: Format adjustment required

-> We have corrected the format

Line 127: Table caption should be placed before the table 

-> We have changed the location of the table caption accordingly

Line 279-280: The authors mentioned that the 1236 DEGs doubled, but which one doubled? Was it in ampicillin-treated group? Please clarify.

-> This means that the greatest difference in any group was selected according to a more than two-fold difference. We have replaced “doubled” with “two-fold” for clarity. (Line 287, in RESULTS)

Line 295-296: “The expression of Bla g 2 RNA more than doubled in the ampicillin-treated group”. Isn’t this ironic compared to the results presented in Figs. 8 and 9? What is the difference between this result not shown and the results presented in Fig. 8? Aren’t they both RNA-seq of the bla gene?

-> We apologize for this error and thank you for your careful review. The indicated sentence was indeed a typographical error, which has been corrected as follows: 

(Lines 375–377, in DISCUSSION) The RNA-seq data showed that the expression level of the Bla g 2 gene (encoding aspartic protease) was reduced by more than four times following ampicillin treatment (S1 Table).

Line 352: Ratio of bacteria to what?

-> We have removed this problematic sentence.

Line 358-359: What did the authors mean by “clustering occurred first among the control group”?

-> This sentence has been modified as follows to clarify the meaning:

(Lines 368–369, in DISCUSSION) …, but samples in the control group clustered well.

Line 369-370: This opening statement is confusing when the authors stated “…to confirm changes in the allergens according to the changes”. Please rephrase this.

-> According to your comment, we have revised the sentences as follows:

(Line 380–381, in DISCUSSION) Therefore, gene expression levels for the major allergens Bla g 1, Bla g 2, and Bla g 5 were measured using qPCR, and the protein production level was also measured.

(Lines 380–381, in DISCUSSION) Similar patterns of expression changes of the major allergens Bla g 1, Bla g 2, and Bla g 5 were observed at both the mRNA and protein levels.

---

## [Editor Report · Decision Letter 1]

3 Nov 2021

PONE-D-21-26586R1Reduced production of the major allergens Bla g 1 and Bla g 2 in Blattella germanica after antibiotic treatmentPLOS ONE

Dear Dr. Yong,

Thank you for submitting your manuscript to PLOS ONE. After careful consideration, we feel that it has merit but does not fully meet PLOS ONE’s publication criteria as it currently stands. Therefore, we invite you to submit a revised version of the manuscript that addresses the points raised during the review process.

We look forward to receiving your revised manuscript.

Kind regards,

Chun Wie Chong

Academic Editor

PLOS ONE

Journal Requirements:

Additional Editor Comments:

Majority of the comments from reviewer 1 have been addressed in the revised version. However, some of the responses to the reviewer's comments were surprisingly left out from the main text. For instance,

1. The rational of using G1 cockroaches

2. The reduction of fertility after ampicillin treatment

3. No morphological differences between the cockroaches

4. The justification to use only sonification for protein extraction

5. The future plan to study the effect or ampicillin treatment to the adult cockroaches

These should be included in the manuscript.

Further, I am not convince that comment 2 from reviewer#2 (However, how does these results relate to the Bla protein production quantified in the study or other insights that might be derived from the results?) had been properly addressed. Also, has the link between Bla protein and microbiome been reported previously? If yes, please elaborate in discussion.

Finally, please provide reference for the the assertion that "Blattabacterium may play a key role in allergen production in cockroaches".
---

## [Author Response · Author response to Decision Letter 1]

5 Nov 2021

Responses to Reviewer Comments

For reviewer 1.

1. The rational of using G1 cockroaches

-> We have included following sentence:

(Lines 106–107 in MATERIALS AND METHODS) We used G1 cockroaches because we had to administer ampicillin immediately after hatching.

2. The reduction of fertility after ampicillin treatment

-> We have included following sentence:

(Lines 233–235 in RESULTS) The number of laid eggs did not vary between the groups, but the offspring were reduced approximately ten times after ampicillin treatment.

3. No morphological differences between the cockroaches

-> We have included following sentence:

(Lines 235–236 in RESULTS) In addition, no morphological differences were observed between the groups.

4. The justification to use only sonification for protein extraction

-> We have included following sentence.

(Lines 150–151 in MATERIALS AND METHODS) Defatting was not performed as there was little fat in the sample and to minimize bacterial contamination.

5. The future plan to study the effect or ampicillin treatment to the adult cockroaches

-> We have included following sentence:

(Lines 428–429 in DISCUSSION) Furthermore, a comparative study on the effect of ampicillin treatment on the microbiome and allergen production between adult and nymph cockroaches is needed.

For reviewer 2.

Further, I am not convince that comment 2 from reviewer#2 (However, how does these results relate to the Bla protein production quantified in the study or other insights that might be derived from the results?) had been properly addressed. Also, has the link between Bla protein and microbiome been reported previously? If yes, please elaborate in discussion.

Finally, please provide reference for the the assertion that "Blattabacterium may play a key role in allergen production in cockroaches".

To the best of our knowledge, the effect of the microbiome on cockroach allergens or that of antibiotic treatment on allergens has not yet been reported. However, in Riptortus pedestris, the absence of the endosymbiont, Burkholderia spp., led to a decrease in vitellogenin (the reference for this study has been added to the manuscript). Vitellogenin was reported as an allergen in B. germanica; however, Burkholderia spp. was not detected in our B. germanica samples.

Based on our results, we inferred that "Blattabacterium may play a key role in allergen production in cockroaches"; therefore, there was no reference provided.

-> The following sentences have been added in the manuscript.

(Lines 408–414 in DISCUSSION) In this study, treatment with ampicillin eliminated Blattabacterium, an endosymbiont of B. germanica, and reduced the production of several allergens. This suggests that Blattabacterium may play a key role in allergen production in cockroaches either directly or indirectly. To the best of our knowledge, the effect of the microbiome on cockroach allergens has not been reported. However, in Riptortus pedestris, the absence of the endosymbiont Burkholderia spp. led to a decrease in vitellogenin, an allergen in B. germanica (26).

---

## [Editor Report · Decision Letter 2]

8 Nov 2021

Reduced production of the major allergens Bla g 1 and Bla g 2 in Blattella germanica after antibiotic treatment

PONE-D-21-26586R2

Dear Dr. Yong,

We’re pleased to inform you that your manuscript has been judged scientifically suitable for publication and will be formally accepted for publication once it meets all outstanding technical requirements.

Kind regards,

Chun Wie Chong

Academic Editor

PLOS ONE
---

## [Editor Report · Acceptance letter]

12 Nov 2021

PONE-D-21-26586R2 

Reduced production of the major allergens Bla g 1 and Bla g 2 in *Blattella germanica* after antibiotic treatment 

Dear Dr. Yong:

I'm pleased to inform you that your manuscript has been deemed suitable for publication in PLOS ONE. Congratulations! Your manuscript is now with our production department. 

Kind regards, 

on behalf of

Dr. Chun Wie Chong 

Academic Editor

PLOS ONE